# Validation of Sepsis-3 using survival analysis and clinical evaluation of quick SOFA, SIRS, and burn-specific SIRS for sepsis in burn patients with suspected infection

**Jaechul Yoon**[1☯]**, Dohern Kym**[1,2☯]**, Jun Hur** [1]*****, **Yong Suk Cho**[1]**, Wook Chun**[1,2]**, Dogeon Yoon**[2]

**1** Department of Surgery and Critical Care, Burn Center, Hangang Sacred Heart Hospital, Hallym University Medical Center, Seoul, Korea, **2** Burn Institute, Hangang Sacred Heart Hospital, Hallym University Medical Center, Seoul, Korea

☯ These authors contributed equally to this work.

* hammerj@hallym.or.kr

## Abstract

### Purpose

Sepsis-3 is a life-threatening organ dysfunction caused by dysregulated host responses to infection; and defined using the Sepsis-3 criteria, introduced in 2016, however, the criteria need to be validated in specific clinical fields. We investigated mortality prediction and compared the diagnostic performance of quick Sequential Organ Failure Assessment (qSOFA), systemic inflammatory response syndrome (SIRS), and burn-specific SIRS (bSIRS) in burn patients.

### Methods

This single-center retrospective cohort study examined burn patients in Seoul, Korea during January 2010–December 2020. Overall, 1,391 patients with suspected infection were divided into four sepsis groups using SOFA, qSOFA, SIRS, and burn-specific SIRS.

### Results

Hazard ratios (HRs) of all unadjusted models were statistically significant; however, the HR (0.726, p = 0.0080.001) in the SIRS $\geq$2 group is below 1. In the adjusted model, HRs of the SOFA $\geq$2 (2.426, <0.001), qSOFA $\geq$2 (7.198, p<0.001), and SIRS $\geq$2 (0.575, p<0.001) groups were significant. The diagnostic performance of dichotomized qSOFA, SIRS, and bSIRS for sepsis was defined by the Sepsis-3 criteria. The mean onset day was 4.13±2.97 according to Sepsis-3. The sensitivity of SIRS (0.989, 95% confidence interval [CI]: 0.982–0.994) was higher than that of qSOFA (0.841, 95% CI: 0.819–0.861) and bSIRS (0.803, 95% CI: 0.779–0.825). Specificities of qSOFA (0.929, 95% CI: 0.876–0.964) and bSIRS (0.922, 95% CI: 0.868–0.959) were higher than those of SIRS (0.461, 95% CI: 0.381–0.543).

**Data Availability Statement:** All data are available from the following DOI: (10.5061/dryad.dbrv15f4d).

**Funding:** This article was supported by a National Research Foundation of Korea (NRF) grant funded by the Korea government (MIST) (Grant No. 2021R1A2C20060331222182102840102). I state 'The funders had no role in study design, data collection and analysis, decision to publish, or preparation of the manuscript.

**Competing interests:** The authors have declared that no competing interests exist.

## Conclusion

Sepsis-3 is a good alternative diagnostic tool because it reflects sepsis severity without delaying diagnosis. SIRS showed higher sensitivity than qSOFA and bSIRS and may therefore more adequately diagnose sepsis.

## Introduction

Sepsis is a systemic response to infection and is one of the most common causes of organ dysfunction and mortality in burn patients and admission to intensive care units (ICUs) among critically ill patients [1, 2]. In particular, the susceptibility for infection or sepsis increases in proportion to the total body surface area (TBSA) burned [3], and the systemic response caused by the burn itself is difficult to distinguish from the systemic inflammatory response syndrome (SIRS) developed during sepsis. Therefore, the diagnosis of sepsis remains challenging, and SIRS and conventional sepsis are not appropriate in burn patients [4, 5]. Therefore, "the Consensus Conference to Define Sepsis and Infection in Burns" was held in Tucson on January 20, 2007, and it proposed alternative definitions of SIRS (bSIRS) and sepsis (i.e., burn sepsis [BS]) [6]. In 2016, Sepsis-3, which discards SIRS and severe sepsis, was redefined using the Sequential Organ Failure Assessment (SOFA) [7]. Many patients with SIRS have no infection, while some patients without SIRS have an infection; SIRS used to diagnose sepsis failed to detect organ dysfunction in one of eight infected patients with organ dysfunction [8, 9]. The quick SOFA (qSOFA) has also been suggested as a bedside screening tool to detect organ dysfunction; however, its use was criticized given the delayed diagnosis that relies on established organ dysfunction, potentially decreasing the sensitivity [10–12]. Since the introduction of Sepsis-3, many studies have validated the new Sepsis-3 and qSOFA in several clinical fields such as emergency departments [13] and ICUs [14] in different medical conditions [15, 16]. However, only few studies have assessed the validation of sepsis-3 in patients with burns. In this study, we investigated the clinical outcome by evaluating the ability of the tools in predicting morality for each sepsis category using survival analyses and assessing the diagnostic performance of SIRS, qSOFA, and bSIRS for sepsis defined by the Sepsis-3 criteria in burn patients with suspected infection.

## Methods

### Study design and population

In this single-center retrospective cohort study, we reviewed electronic medical records of patients with suspected infection from January 2010 to December 2020. The patients were aged over 18 years, were admitted to the Burn Intensive Care Unit (BICU) of our institution within 2 days after the burns, and had undergone acute fluid resuscitation during the first 2 days of admission. Overall, 2384 patients aged over 18 years were included, and we excluded 440 patients because they were burned 24 hours before admission, 185 owing to incomplete fluid resuscitation, and 368 owing to no-suspected infection. In total, 1,391 eligible patients were divided into four subgroups according to different criteria (Fig 1). All analyzed patients were primarily admitted as they had burns and a suspected infection during BICU admission. This study was approved by the Institutional Review Board of Hangang Sacred Heart Hospital (HG2017-063). The requirement for informed consent was waived owing to the retrospective nature of the study.

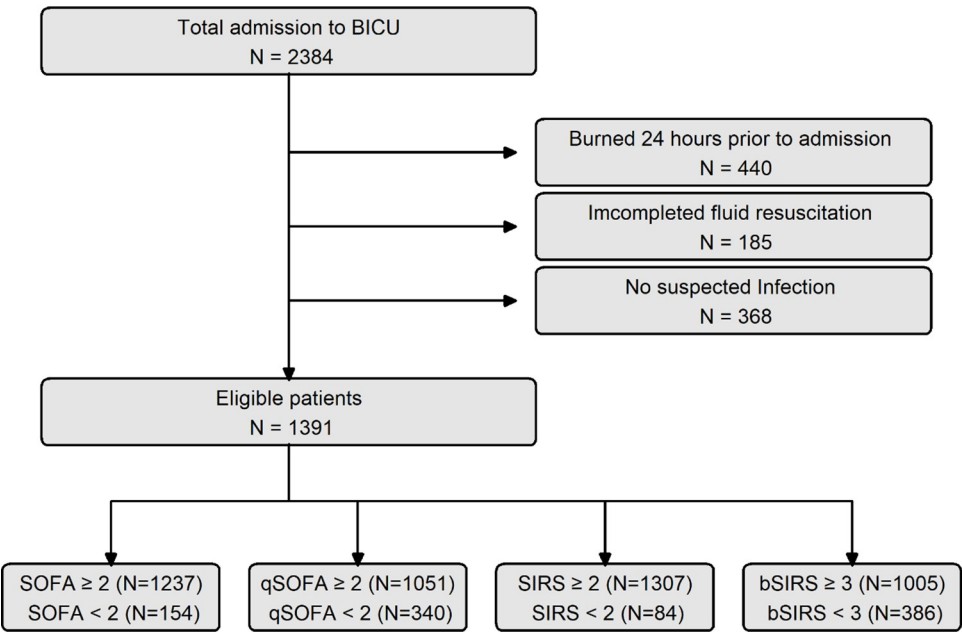

**Fig 1. Flowchart of enrolled patients for the survival analysis in this study.**

## Data collection and missing values

The electrical medical records were anonymized and retrieved from the clinical database warehouse in Hangang Sacred Heart Hospital, Hallym University, Seoul, Korea. The database consists of structured data collected prospectively for all patients admitted to the BICU. Patient demographics such as age, sex, comorbidities, and variables for burn-specific scoring systems such as the Abbreviated Burn Severity Index (ABSI) [17], Hangang [18], and revised Baux (rBaux) [19] scores were calculated for each patient upon admission. For critically ill patients, additional scores, such as the Acute Physiology and Chronic Health Evaluation (APACHE) IV, were calculated [20]. The proportion of missing data at admission was <5% for all variables. These missing values reflected tests not being performed at the physicians' discretion under the assumption that they would return normal results; therefore, these missing values were replaced by normal values in the analyses. The scores of SOFA, qSOFA, SIRS, and bSIRS were calculated daily. Missing data regarding longitudinal variables were replaced with the values at the last observation. Only four longitudinal biomarkers (prothrombin time, lactate, total bilirubin, and bicarbonate) had more than 30% missing data. If the measured data were collected several times per day, the worst value of the day was included. The proportions and patterns of missing baseline and daily data are shown in S1 and S2 Figs in S1 File.

## Outcomes and definitions

The primary outcome was 60-day in-ICU mortality. The secondary outcome was ICU length of stay (LOS). The gold standard for the diagnosis of sepsis (i.e., S3) was defined as an acute change in total SOFA scores of ≥2 points according to the Sepsis-3 criteria. Sepsis-3q (S3q) was defined as a qSOFA score ≥2, consisting of an altered mental state, respiratory rate ≥22 breaths/min, and systolic blood pressure ≤100 mmHg [7]. Sepsis-1 (S1) was defined as a SIRS score ≥2, consisting of a temperature >38°C or <36°C, heart rate >90 beats/min, respiratory rate >20 breaths/min, and white blood cell count >12,000 cells/mm$^3$, <4,000/mm$^3$, or >10%

immature (band) forms. BS was defined as bSIRS ≥3, consisting of (1) a temperature >39˚C or <36.5˚C, (2) progressive tachycardia >110 bpm, (3) progressive tachypnea >25 bpm (not venti-lated) or with minute ventilation >12 L/min, (4) thrombocytopenia <100,000 cells/$\mu$l, (5) hyper-glycemia (in the absence of pre-existing diabetes mellitus) as indicated by an untreated plasma glucose >200 mg/dl, or by the presence of insulin resistance, and (6) inability to continue enteral feedings over 24 hours [6]. A suspected infection was defined as a situation where cultures to determine the infection source (i.e., blood, wounds, sputum, urine) were performed and antibiot-ics were administered before the results were received. A documented infection was defined as the presence of positive culture infection, pathologic tissue source, or clinical response to antimi-crobials. Four sepsis criteria were assessed on a daily basis throughout the study period.

## Statistical analysis

All continuous variables are expressed as means ± SD for normal distributions or median and interquartile ranges (IQR) for non-normal distributions. The independent t-test or Wilcoxon signed-rank test was used to determine differences between the two groups. Categorical vari-ables were analyzed using the chi-squared test, and the frequencies are presented as percent-ages. The differences in the onset day for each sepsis category were compared using paired t-tests, with the onset day of each category for Sepsis-3 as the reference. Accuracy, sensitivity, specificity, positive predictive value, and negative predictive value for the diagnosis of sepsis were calculated. The hazard ratios (HRs) of in-ICU mortality for four dichotomized groups (i.e., S3, S3q, S1, and BS) were estimated using an extended Cox proportional hazards model to obtain longitudinal insight into how the individual components of the model affected the overall mortality of patients [21]. HR was also calculated after adjusting for known predictors of burn severity such as age, TBSA burned, and inhalation injury. Harrell's C-index was used to evaluate the prediction accuracy of the models. P <0.05 (two-sided) was considered to be statistically significant. Data management and statistical analysis were performed using R (A language and environment for statistical computing, Vienna, Austria, Version 4.1.0).

## Results

### Study population

In this study, 1,391 burn patients with a suspected infection between January 2010 and Decem-ber 2021 were included. All patients were divided into four groups according to the sepsis cri-teria. In total, 1,237 patients were diagnosed as having Sepsis-3 (S3), 1,307 patients as having Sepsis-1 (S1), 1,051 patients as having Sepsis-3q (S3q), and 1,005 patients as having BS (Fig 1 and Table 1).

Among them, 329 patients (23.73%) died. The overall median age was 50 years with a male predominance (81%). The median TBSA was 32.0%, and 649 patients (47%) presented inhala-tion injury. The median LOS in the ICU was 19.0 days. The median (interquartile) ABSI, rBaux, Hangang, and APACHE IV scores were 8 (7–10), 93 (76–114), 133 (122–148), and 38 (25–55), respectively. The infection sources of the sepsis were mostly the sputum (40%), wounds (35%), urine (23%), and blood (17%) (Table 2).

### Survival analysis according to the sepsis criteria

All survival curves showed statistically significant differences in the sepsis criteria between the groups using log-rank tests. All HRs in the unadjusted models were statistically significant; however, the HR (0.726, p = 0.008) in patients diagnosed using Sepsis-1 was lower than that in patients diagnosed using other sepsis criteria. In the adjusted model, the HRs of Sepsis-3

**Table 1. Demographics and infection sources according to sepsis categories.**

| Variables | Sepsis-3, N = 1,237 | Sepsis-3q, N = 1,051 | Sepsis-1, N = 1,307 | Burn Sepsis, N = 1,005 | p-value |
|---|---|---|---|---|---|
| **Demographics** | | | | | |
| Mortality | | | | | <0.001 |
| Survivors | 919 (74.3%) | 736 (70.0%) | 991 (75.8%) | 694 (69.1%) | |
| Non-survivors | 318 (25.7%) | 315 (30.0%) | 316 (24.2%) | 311 (30.9%) | |
| Detection Days | | | | | <0.001 |
| Mean (SD) | 4.20 (2.80) | 6.33 (4.82) | 4.22 (2.63) | 5.37 (4.74) | |
| Patient Age (years) | | | | | 0.429 |
| Median [IQR] (Range) | 51 [41, 60] (18–99) | 51 [42, 61] (18–99) | 50 [41, 60] (18–99) | 51 [42, 61] (18–99) | |
| Sex | | | | | 0.775 |
| Female | 243 (20%) | 223 (21%) | 258 (20%) | 200 (20%) | |
| Male | 994 (80%) | 828 (79%) | 1,049 (80%) | 805 (80%) | |
| Type | | | | | 0.829 |
| ChB | 15 (1.2%) | 13 (1.2%) | 18 (1.4%) | 14 (1.4%) | |
| CoB | 38 (3.1%) | 33 (3.1%) | 40 (3.1%) | 34 (3.4%) | |
| EB | 132 (11%) | 92 (8.8%) | 147 (11%) | 88 (8.8%) | |
| FB | 958 (77%) | 828 (79%) | 996 (76%) | 783 (78%) | |
| SB | 94 (7.6%) | 85 (8.1%) | 106 (8.1%) | 86 (8.6%) | |
| TBSA | | | | | <0.001 |
| Median [IQR] (Range) | 35 [21, 55] (0–99) | 38 [24, 60] (0–99) | 33 [20, 52] (0–99) | 40 [25, 60] (0–99) | |
| Inhalation | | | | | 0.427 |
| No | 636 (51%) | 519 (49%) | 686 (52%) | 502 (50%) | |
| Yes | 601 (49%) | 532 (51%) | 621 (48%) | 503 (50%) | |
| LOICU | | | | | <0.001 |
| Median [IQR] (Range) | 21 [11, 36] (4–60) | 24 [13, 38] (4–60) | 20 [10, 35] (4–60) | 24 [13, 39] (4–60) | |
| **Severity Scores** | | | | | |
| ABSI | | | | | <0.001 |
| Median [IQR] (Range) | 9 [7, 11] (3–16) | 9 [7, 11] (4–16) | 8 [7, 10] (3–16) | 9 [7, 11] (4–16) | |
| rBaux | | | | | <0.001 |
| Median [IQR] (Range) | 96 [79, 116] (30–187) | 99 [83, 119] (38–187) | 94 [77, 115] (24–187) | 100 [84, 120] (24–187) | |
| Hangang | | | | | <0.001 |
| Median [IQR] (Range) | 135 [124, 150] (94–209) | 138 [126, 152] (97–209) | 134 [123, 148] (94–209) | 139 [127, 153] (99–209) | |
| APACHE_IV | | | | | <0.001 |
| Median [IQR] (Range) | 40 [28, 56] (1–124) | 42 [30, 58] (3–124) | 39 [26, 55] (1–124) | 43 [30, 59] (3–124) | |
| **Infection Sources** | | | | | |
| Blood | | | | | 0.016 |
| Negative | 459 (75%) | 430 (68%) | 461 (74%) | 409 (70%) | |
| Positive | 155 (25%) | 205 (32%) | 161 (26%) | 175 (30%) | |
| Unknown | 623 | 416 | 685 | 421 | |
| Wound | | | | | 0.003 |
| Negative | 81 (18%) | 56 (11%) | 84 (18%) | 54 (12%) | |
| Positive | 371 (82%) | 446 (89%) | 391 (82%) | 385 (88%) | |
| Unknown | 785 | 549 | 832 | 566 | |
| Sputum | | | | | 0.025 |
| Negative | 111 (16%) | 71 (11%) | 111 (15%) | 75 (12%) | |
| Positive | 605 (84%) | 585 (89%) | 627 (85%) | 553 (88%) | |
| Unknown | 521 | 395 | 569 | 377 | |
| Urine | | | | | 0.719 |

(*Continued*)

**Table 1.** (Continued)

| Variables | Sepsis-3, N = 1,237 | Sepsis-3q, N = 1,051 | Sepsis-1, N = 1,307 | Burn Sepsis, N = 1,005 | p-value |
|---|---|---|---|---|---|
| Negative | 447 (59%) | 427 (60%) | 459 (57%) | 393 (59%) | |
| Positive | 317 (41%) | 290 (40%) | 349 (43%) | 273 (41%) | |
| Unknown | 473 | 334 | 499 | 339 | |
| Line Tip | | | | | 0.145 |
| Negative | 74 (70%) | 89 (61%) | 76 (72%) | 70 (62%) | |
| Positive | 31 (30%) | 58 (39%) | 29 (28%) | 42 (38%) | |
| Unknown | 1,132 | 904 | 1,202 | 893 | |
| Pus | | | | | >0.999 |
| Negative | 1 (33%) | 1 (17%) | 1 (33%) | 0 (0%) | |
| Positive | 2 (67%) | 5 (83%) | 2 (67%) | 1 (100%) | |
| Unknown | 1,234 | 1,045 | 1,304 | 1,004 | |
| Other | | | | | 0.548 |
| Negative | 10 (16%) | 6 (10%) | 11 (15%) | 5 (8.8%) | |
| Positive | 54 (84%) | 53 (90%) | 60 (85%) | 52 (91%) | |
| Unknown | 1,173 | 992 | 1,236 | 948 | |

(2.426, <0.001), Sepsis-3q (7.198, p < 0.001), and Sepsis-1 (0.575, p < 0.001) were statistically significant, with the latter showing a negative prediction. The HR of BS (0.966, p = 0.775) was below 1 and was not statistically significant. Adjusting its HR for known predictors such as age, TBSA, and inhalation injury, lowered its value, indicating that BS may be affected by burn factors but not by sepsis itself (Fig 2; S1 Table in S1 File). The mortality in each sepsis subgroup was 30.0% in BS, 30.0% in S3q, 25.7% in S3, and 24.2% in S1 (S3 Fig in S1 File).

### Detection days of each sepsis criteria

We analyzed the first detection day for each sepsis criteria to evaluate whether Sepsis-3 could diagnose sepsis early in the 909 patients who met all four sepsis criteria. The intersection of the four sepsis categories is presented in S4 Fig in S1 File. The mean onset day was the lowest with 4.13 ± 2.97 in Sepsis-3 and was statistically different with the onset in Sepsis-1 (4.23 ± 2.83), Sepsis-3q (6.26 ± 4.76), and BS (5.36 ± 4.88), as shown in S2 Table in S1 File; Fig 3.

### Diagnostic performance comparison of qSOFA, SIRS, and bSIRS for Sepsis-3

We evaluated the diagnostic performance of dichotomized qSOFA, SIRS, and bSIRS for sepsis defined by the Sepsis-3 criteria. The flow chart of the three divided groups is shown in S5 Fig in S1 File. The sensitivity of SIRS (0.989, 95% CI: 0.982–0.994) was higher than that of qSOFA (0.841, 95% CI: 0.819–0.861) and bSIRS (0.803, 95% CI: 0.779–0.825). The specificity of qSOFA (0.929, 95% CI: 0.876–0.964) and bSIRS (0.922, 95% CI: 0.868–0.959) was higher than that of SIRS (0.461, 95% CI: 0.381–0.543) (S3 Table in S1 File). The diagnostic performance of these criteria and the Venn diagram are presented in Fig 4.

## Discussion

### Key findings

This large cohort study of burned patients with suspected infection from South Korea showed the clinical utility of the Sepsis-3 criteria to diagnose sepsis and distinguish the systemic

**Table 2. Demographic and infection sources according to 60-day mortality.**

| Group | Variables | Overall, N = 1,391 | Survivors, N = 1,062 | Non-survivors, N = 329 | p-value |
|---|---|---|---|---|---|
| Demographics | Patient Age | | | | <0.001 |
| | Median [IQR] (Range) (years) | 50 [41, 60] (18–99) | 49 [40, 58] (18–99) | 54 [45, 65] (18–96) | |
| | Sex | | | | 0.704 |
| | Female | 269 (19%) | 203 (19%) | 66 (20%) | |
| | Male | 1,122 (81%) | 859 (81%) | 263 (80%) | |
| | Type | | | | |
| | ChB | 20 (1.4%) | 17 (1.6%) | 3 (0.9%) | |
| | CoB | 43 (3.1%) | 35 (3.3%) | 8 (2.4%) | |
| | EB | 166 (12%) | 156 (15%) | 10 (3.0%) | |
| | FB | 1,048 (75%) | 764 (72%) | 284 (86%) | |
| | SB | 114 (8.2%) | 90 (8.5%) | 24 (7.3%) | |
| | TBSA | | | | <0.001 |
| | Median [IQR] (Range) | 32 [20, 51] (0–99) | 27 [18, 40] (0–91) | 62 [40, 81] (4–99) | |
| | Inhalation | | | | <0.001 |
| | No | 742 (53%) | 616 (58%) | 126 (38%) | |
| | Yes | 649 (47%) | 446 (42%) | 203 (62%) | |
| | LOICU | | | | <0.001 |
| | Median [IQR] (Range) | 19 [9, 34] (2–60) | 21 [8, 37] (2–60) | 15 [10, 24] (2–58) | |
| Severity Scores | ABSI | | | | <0.001 |
| | Median [IQR] (Range) | 8 [7, 10] (3–16) | 8 [6, 9] (3–14) | 12 [10, 13] (5–16) | |
| | rBaux | | | | <0.001 |
| | Median [IQR] (Range) | 93 [76, 114] (24–187) | 86 [70, 100] (24–153) | 127 [108, 145] (54–187) | |
| | Hangang | | | | <0.001 |
| | Median [IQR] (Range) | 133 [122, 148] (94–209) | 128 [119, 138] (94–184) | 159 [149, 173] (119–209) | |
| | APACHE_IV | | | | <0.001 |
| | Median [IQR] (Range) | 38 [25, 55] (1–124) | 33 [22, 46] (1–96) | 59 [45, 75] (17–124) | |
| | SOFA | | | | <0.001 |
| | Median [IQR] (Range) | 4 [3, 6] (0–16) | 4 [3, 5] (0–12) | 7 [5, 9] (2–16) | |
| Comorbidities | Hypertension | | | | <0.001 |
| | No | 1,132 (81%) | 886 (83%) | 246 (75%) | |
| | Yes | 259 (19%) | 176 (17%) | 83 (25%) | |
| | DM | | | | 0.001 |
| | No | 1,277 (92%) | 989 (93%) | 288 (88%) | |
| | Yes | 114 (8.2%) | 73 (6.9%) | 41 (12%) | |
| | Tuberculosis | | | | 0.521 |
| | No | 1,367 (98%) | 1,045 (98%) | 322 (98%) | |
| | Yes | 24 (1.7%) | 17 (1.6%) | 7 (2.1%) | |
| | Hepatobiliary | | | | 0.363 |
| | No | 1,361 (98%) | 1,037 (98%) | 324 (98%) | |
| | Yes | 30 (2.2%) | 25 (2.4%) | 5 (1.5%) | |
| | Cardiovascular | | | | 0.569 |
| | No | 1,360 (98%) | 1,037 (98%) | 323 (98%) | |
| | Yes | 31 (2.2%) | 25 (2.4%) | 6 (1.8%) | |
| | CVA | | | | >0.999 |
| | No | 1,371 (99%) | 1,046 (98%) | 325 (99%) | |
| | Yes | 20 (1.4%) | 16 (1.5%) | 4 (1.2%) | |
| | Cancer | | | | 0.460 |

*(Continued)*

**Table 2.** (Continued)

| Group | Variables | Overall, N = 1,391 | Survivors, N = 1,062 | Non-survivors, N = 329 | p-value |
|---|---|---|---|---|---|
| | No | 1,364 (98%) | 1,043 (98%) | 321 (98%) | |
| | Yes | 27 (1.9%) | 19 (1.8%) | 8 (2.4%) | |
| | Hyperlipidemia | | | | 0.795 |
| | No | 1,350 (97%) | 1,030 (97%) | 320 (97%) | |
| | Yes | 41 (2.9%) | 32 (3.0%) | 9 (2.7%) | |
| | Other | | | | 0.068 |
| | No | 1,051 (76%) | 790 (74%) | 261 (79%) | |
| | Yes | 340 (24%) | 272 (26%) | 68 (21%) | |
| Sepsis categories | Sepsis by SOFA $\geq$2 | 1,237 (89%) | 919 (87%) | 318 (97%) | <0.001 |
| | Sepsis by qSOFA $\geq$2 | 1,051 (76%) | 736 (69%) | 315 (96%) | <0.001 |
| | Sepsis by SIRS $\geq$2 | 1,307 (94%) | 991 (93%) | 316 (96%) | 0.069 |
| | Sepsis by bSIRS $\geq$3 | 1,005 (72%) | 694 (65%) | 311 (95%) | <0.001 |
| Infection Sources | Blood | | | | <0.001 |
| | Negative | 1,159 (83%) | 918 (86%) | 241 (73%) | |
| | Positive | 232 (17%) | 144 (14%) | 88 (27%) | |
| | Wound | | | | 0.011 |
| | Negative | 906 (65%) | 711 (67%) | 195 (59%) | |
| | Positive | 485 (35%) | 351 (33%) | 134 (41%) | |
| | Sputum | | | | 0.001 |
| | Negative | 828 (60%) | 657 (62%) | 171 (52%) | |
| | Positive | 563 (40%) | 405 (38%) | 158 (48%) | |
| | Urine | | | | 0.022 |
| | Negative | 1,071 (77%) | 833 (78%) | 238 (72%) | |
| | Positive | 320 (23%) | 229 (22%) | 91 (28%) | |
| | Line Tip | | | | 0.035 |
| | Negative | 1,317 (95%) | 1,013 (95%) | 304 (92%) | |
| | Positive | 74 (5.3%) | 49 (4.6%) | 25 (7.6%) | |
| | Pus | | | | >0.999 |
| | Negative | 1,385 (100%) | 1,057 (100%) | 328 (100%) | |
| | Positive | 6 (0.4%) | 5 (0.5%) | 1 (0.3%) | |
| | Other | | | | 0.436 |
| | Negative | 1,322 (95%) | 1,012 (95%) | 310 (94%) | |
| | Positive | 69 (5.0%) | 50 (4.7%) | 19 (5.8%) | |

response induced by sepsis from that induced by burns themselves. The Sepsis-3 criteria can predict mortality even when adjusted for known burn predictors without delaying diagnosis. The Sepsis-1 criteria can diagnose sepsis similar to the Sepsis-3 criteria in the early days; however, it can predict the mortality reversely with an adjusted HR of 0.646. The qSOFA and bSIRS criteria showed high specificity and low sensitivity; therefore, limiting their utility as diagnostic tools. There is no need to use diagnostic criteria specific to burn patients in this cohort; thus, the Sepsis-3 criteria can be used to compare them with patients with other diseases associated with sepsis. Sepsis-3, defined by SOFA $\geq$2, can be useful as an alternative diagnostic tool.

## Relationship to previous studies

We previously reported a somewhat different conclusion. In our previous study, the Sepsis-3 criteria were not superior to the other BS criteria in the present study [22]. This is because the

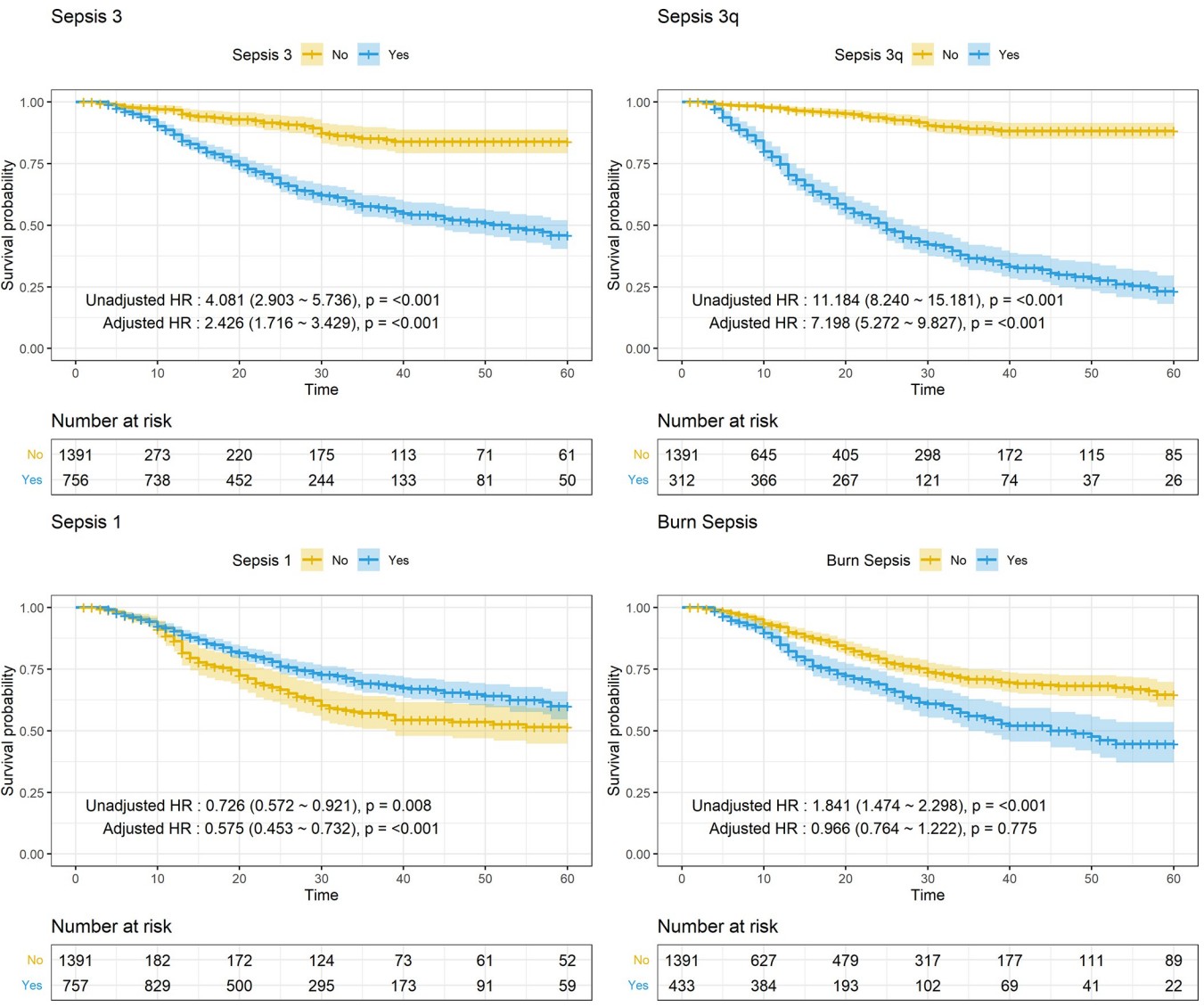

**Fig 2. Survival curves (Kaplan-Meier) and unadjusted/adjusted HR of each sepsis criterion related to the 60-day mortality in burned patients with suspected infection.** (A) Sepsis-3. (B) Sepsis-3q. (C) Sepsis-1. (D) Burn Sepsis.

performance of mortality prediction was evaluated cross-sectionally at the time of diagnosis. The evaluation of the cut-off value of SOFA as a Sepsis-3 criterion might be limited as a diagnostic tool. Burns and trauma are two well-known conditions in which SIRS develops without infections, implying that that SIRS may not immediately detect sepsis in this population. Therefore, the diagnosis of sepsis in burn patients is complicated by the inability to distinguish the systemic response induced by infection and the burn itself. Burn physicians ignore SIRS as it is considered to be a normal response. Due to this, there are cases in which patients with sepsis are missed, although the actual sepsis is progressing [23]. Therefore, the modified definition of SIRS is proposed by ABA and is determined by meeting at least three variables among the following: temperature, heart rate, respiration rate, platelet count, blood glucose, and intolerance of feeding [6]. Nevertheless, several studies have reported the best indicators of sepsis in thermally injured patients [24]. Yan et al. reported that Sepsis-3 showed higher sensitivity

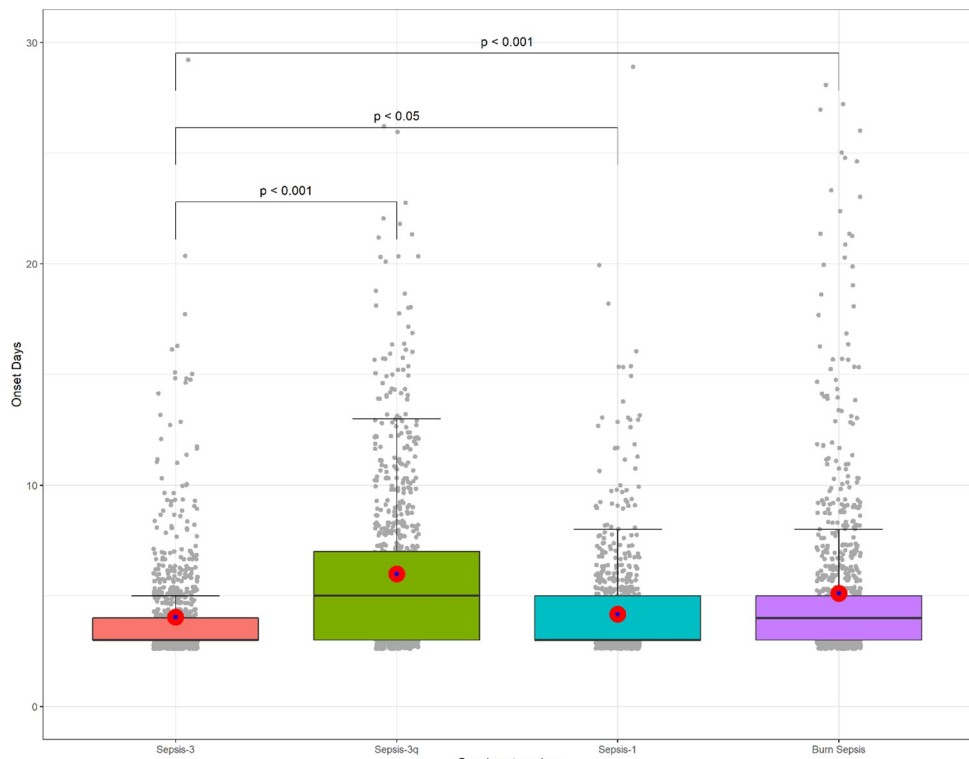

**Fig 3. Boxplot of onset days for the four categories of sepsis (gray dots = sample data points, black dots = outlier, blue dots = mean, red dots = 95% confidence interval).**

(85%) than BS (59%), defined by ABA, and that multiple organ dysfunction is more specific for sepsis than inflammation. However, their dataset did not collect all relevant biomarker data prospectively, and their study included a smaller cohort than our own [4]. In this study, we found that the SIRS is too sensitive with a low specificity to detect sepsis. SIRS could not stratify the severity of sepsis based on our results that the HR ratio is lower than 1. This showed a negative prediction for the group, in which sepsis was defined by SIRS $\geq$2. bSIRS has high specificity and lower sensitivity, which reflects the severity but is not an appropriate diagnostic tool. Additionally, it showed adjusted HR values of 0.966, indicating that it cannot stratify the risk of mortality and could be affected by burn factors but not sepsis itself. qSOFA showed similar sensitivity and specificity to bSIRS but not to bSIRS; its HR showed a better discriminative power for mortality prediction. The specificity of qSOFA was higher than that of SIRS in our study and was comparable to that reported in other studies [15, 25].

## Strengths and limitations

This study represents the largest series ever reported in the literature in this population. Although this was a retrospective study, the data were prospectively collected and stored in an electronic data warehouse. Our study is the first evaluation of qSOFA in burn patients with suspected infection, and it showed that qSOFA had similar clinical sensitivity and specificity as bSIRS. It also used an extended Cox analysis to determine the predictive ability of daily-collected SIRS, SOFA, qSOFA, and bSIRS scores, as almost always, the infection and sepsis-like clinical symptoms occur in burn patients throughout ICU hospitalization. We also evaluated the onset days of each sepsis category in the same cohort to determine whether diagnosis was delayed. However, this study had several limitations. First, this was a single-center study that

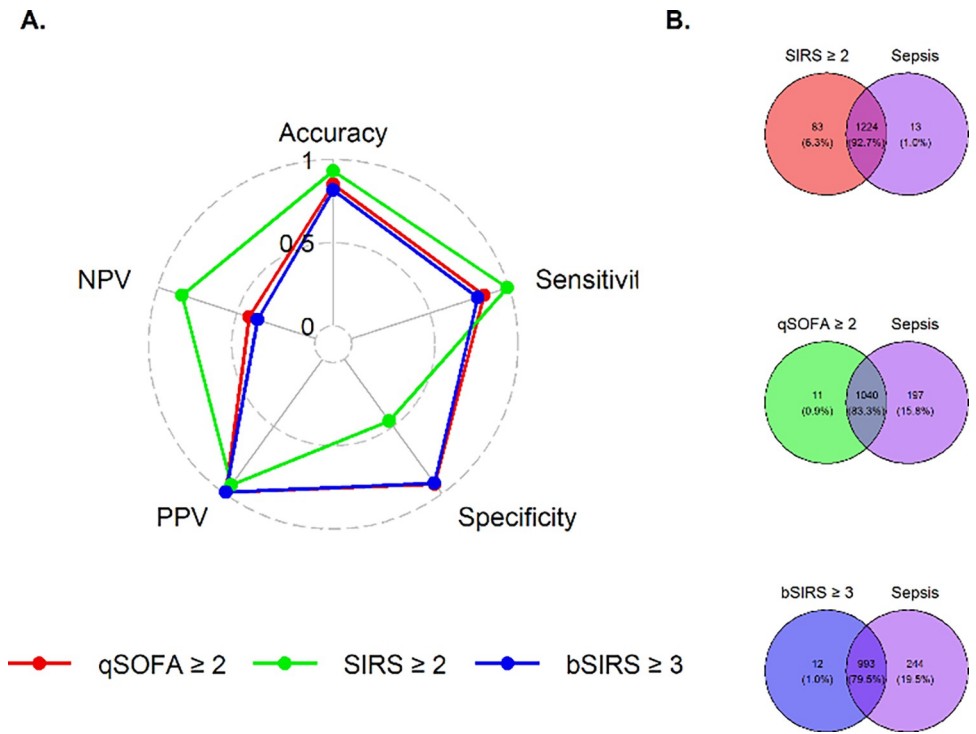

**Fig 4. Comparison of the diagnostic performance of qSOFA, SIRS, and bSIRS for sepsis.** (A) Diagnostic performance. (B) Venn diagrams.

reflects the specific case-mix of our center, and we recognize that the results will depend on our population characteristics and their specific treatments. Second, qSOFA was introduced as a bedside screening tool, but it was evaluated as a diagnostic tool for sepsis in this study. Third, missing data were considered to be normal or previous values. Fourth, regarding the definition of a suspected infection, it is assumed that the culture was undertaken when the physician suspected an infection; however, it cannot be ascertained whether this was the actual time of suspicion or routine tests. Fifth, the primary outcome in this study is mortality, which cannot be used to definitively validate the ability to predict sepsis. However, given that it is very difficult to detect the systemic response associated with sepsis, numerous physicians simply refer to "septic" patients, rather than strictly defined "sepsis." Therefore, prediction of mortality, which is a crucial indicator in critical care patients, is an indirect way of evaluating the efficacy of a diagnostic tool for sepsis [14]. Furthermore, several studies have reported the validation of sepsis diagnostic tools using mortality as the primary outcome [15].

## Conclusion

Sepsis defined by SOFA ≥2 can be a good alternative diagnostic tool because it reflects the severity of sepsis well using the survival analysis. SIRS showed higher sensitivity and lower specificity, while qSOFA and bSIRS showed higher specificity and lower sensitivity for the diagnosis of sepsis; therefore, they are not adequate for the diagnosis of sepsis.

## Supporting information

**S1 File. Supplementary and further details.**
(DOCX)

## Author Contributions

**Conceptualization:** Dohern Kym, Jun Hur, Yong Suk Cho.

**Data curation:** Dohern Kym.

**Investigation:** Jun Hur.

**Methodology:** Dogeon Yoon.

**Project administration:** Jaechul Yoon, Yong Suk Cho, Dogeon Yoon.

**Resources:** Wook Chun.

**Supervision:** Yong Suk Cho, Wook Chun.

**Validation:** Dohern Kym, Yong Suk Cho.

**Visualization:** Jun Hur.

**Writing – original draft:** Jaechul Yoon.

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
