## [Decision Letter · Decision Letter 0]

1 Jul 2022

PONE-D-22-15515Validation of Sepsis-3 using survival analysis and clinical evaluation of quick SOFA, SIRS, burn-specific SIRS for sepsis in burn patients with suspected infection.PLOS ONE

Dear Dr. Hur,

Thank you for submitting your manuscript to PLOS ONE. After careful consideration, we feel that it has merit but does not fully meet PLOS ONE’s publication criteria as it currently stands. Therefore, we invite you to submit a revised version of the manuscript that addresses the points raised during the review process.

We have sent your manuscript to three expert reviewers. While they believe your study has merit, they have brought up significant concerns that need to be addressed. 

While all had good suggestions that should be responded to, Reviewer 2's points 2 and 3 were the most pressing to be addressed. We hope you decide to submit a revised manuscript to PLoS One.

We look forward to receiving your revised manuscript.

Kind regards,

David M. Burmeister, PhD

Academic Editor

PLOS ONE

Journal Requirements:

Additional Editor Comments:

Dear Dr. Hur,

Thank you for your submission to PLoS One. We have sent your manuscript to three expert reviewers. While they believe your study has merit, they have brought up significant concerns that need to be addressed. While all had good suggestions that should be responded to, Reviewer 2's points 2 and 3 were the most pressing to be addressed.

We hope you decide to submit a revised manuscript to PLoS One.

Reviewers' comments:

Reviewer's Responses to Questions

**Comments to the Author**

1. Is the manuscript technically sound, and do the data support the conclusions?

Reviewer #1: Yes

Reviewer #2: No

Reviewer #3: Yes

2. Has the statistical analysis been performed appropriately and rigorously? 

Reviewer #1: Yes

Reviewer #2: No

Reviewer #3: Yes

3. Have the authors made all data underlying the findings in their manuscript fully available?

Reviewer #1: Yes

Reviewer #2: Yes

Reviewer #3: Yes

4. Is the manuscript presented in an intelligible fashion and written in standard English?

Reviewer #1: Yes

Reviewer #2: Yes

Reviewer #3: Yes

5. Review Comments to the Author

Reviewer #1: I think this is an interesting and important study. My questions are

how do you know that sepsis 3 is best using just stats models? any prospective data collection?

now that you know sepsis 3 is best, how can you use this info to improve outcomes?

do you have info on what type of bacteria?

your table and figures and table need labeling of significances.

plse discuss what is novel about your findings as sepsis 3 was deemed best in burns in a study by yan et al.

Reviewer #2: This paper reviews three different sepsis definitions: quick Sequential Organ Failure Assessment (qSOFA),

systemic inflammatory response syndrome (SIRS), and burn-specific SIRS (bSIRS) in burn patients for their ability to predict mortality in burn patients. They conclude that Sepsis-3 is a good measure for predicting sepsis. The paper is clearly written but does need some editing for syntax and grammar. There are several fundamental issues with the analyses.

1. No definition for infection or diagnosis of infection is provided in the manuscript. This needs to be added to assure diagnostic consistency.

2. The primary outcome for this study is mortality. This does not validate the efficacy of a sepsis prediction system. The paper needs to be modified to reflect this differential.

3. Missing values were estimated using several different methodologies. No information is given with respect to the number of missing values or how the analysis would be different when run with only existing values. This needs to be added to the paper, as imputation of values introduces bias.

4. The sepsis markers are measured daily throughout admission rather than at a fixed time point. How can we use such a system to predict sepsis?

Reviewer #3: Gold standard definition for sepsis noted as acute change in SOFA scores, please provide a cross tab table of patients meeting the Sepsis definition (SIRS, SEPSIS-3, etc) vs actual concurrent culture positivity. That is to say I'm not surprised large burns all seem to have at least one infectious complication, but by table 1 alone and the accompanying results I'm not convinced of the temporal relation of these findings. Figure 3 makes a suggestion that there is a temporal relationship please expand and state explicitly.

Additionally in a cohort of burn patients with a median TBSA of 32 I would expect multiple operative visits. Each visit can cause a systemic response similar to sepsis, how was this controlled for in the study. Is there a temporal relationship to the development of the above criteria linked to operative intervention. Does that hold durable to predict sepsis in burn.

All in a very important and timely paper.

6. PLOS authors have the option to publish the peer review history of their article (what does this mean?). If published, this will include your full peer review and any attached files.

Reviewer #1: No

Reviewer #2: No

Reviewer #3: No

---

## [Author Response · Author response to Decision Letter 0]

1 Sep 2022

July 12, 2022

PLOS One

Dear Editor:

Thank you again for providing us with the opportunity to revise our manuscript. We have enclosed a summary of the changes made based on the valuable comments provided by the reviewers. All of the issues raised have been addressed, and the changes are marked in red font in the revised manuscript, which has been uploaded as a separate file labeled “Revised Manuscript with Tracked Changes.”

We thank you and the reviewers for your thoughtful suggestions and insights, which have enriched the manuscript and produced a better and more balanced account of the research. We hope that the revised manuscript is now suitable for publication in your journal.

Sincerely,

Jun Hur

Department of Surgery and Critical Care 

Burn Center 

Hangang Sacred Heart Hospital 

Hallym University Medical Center

12, Beodeunaru-ro 7-gil, Youngdeungpo-gu, Seoul, Korea, 07247

Tel. 82-2-2639-5446 

Fax. 82-2-2678-4386 

E-mail: hammerj@hallym.or.kr

Reviewer #1: I think this is an interesting and important study. 

My questions are

how do you know that sepsis 3 is best using just stats models? any prospective data collection?

Response: Our data were anonymized and retrieved from a prospectively collected clinical database. Sepsis-3 was defined using this prospectively collected dataset. Furthermore, our laboratory data are longitudinal and time-series data, not cross-sectional data. We performed an extended Cox proportional hazards analysis to obtain longitudinal insight into how individual components of the model affected overall mortality. Therefore, we believe that our statsmodels can indeed show that Sepsis-3 is superior to other definitions.

now that you know sepsis 3 is best, how can you use this info to improve outcomes?

Response: Our results showed that Sepsis-3 was detected earlier than the other sepsis criteria. This finding would be crucial in allowing physicians to make prompt, early decisions without sacrificing sensitivity and specificity. Early detection of sepsis can help improve outcomes. 

do you have info on what type of bacteria?

Response: We only checked whether or not cultures were performed to define a suspected infection; unfortunately, we did not collect data on the type of bacteria. We plan to conduct a future study on bacterial profiles of burn patients with sepsis. We regret that we cannot provide these data now but hope that you will look forward to our next study.

your table and figures and table need labeling of significances.

Response: We have provided the p-values in the table and figures as per your comment.

plse discuss what is novel about your findings as sepsis 3 was deemed best in burns in a study by yan et al.

Response: Yan et al. proposed that a system for prospectively and routinely documenting all relevant predictors for each sepsis criterion would provide data that allow for a more detailed analysis of each criterion. This approach would overcome the limitations of Yan et al.’s study. Our study involved the routine prospective collection of a longitudinal dataset from the largest cohort reported in the field of burns surgery and confirmed that Sepsis-3 is indeed the optimal criterion. Therefore, we believe that our study is indeed novel. We have emphasized this point in the revised Discussion section.

Reviewer #2: This paper reviews three different sepsis definitions: quick Sequential Organ Failure Assessment (qSOFA), systemic inflammatory response syndrome (SIRS), and burn-specific SIRS (bSIRS) in burn patients for their ability to predict mortality in burn patients. They conclude that Sepsis-3 is a good measure for predicting sepsis. The paper is clearly written but does need some editing for syntax and grammar. There are several fundamental issues with the analyses.

1. No definition for infection or diagnosis of infection is provided in the manuscript. This needs to be added to assure diagnostic consistency.

Response: A suspected infection was defined as a situation where a culture was performed and antibacterial agents were administered before the culture results were obtained. A documented infection was defined as the presence of positive culture infection, pathologic tissue source, or clinical response to antimicrobials. We have clarified this in the revised Outcomes and Definitions subsection.

2. The primary outcome for this study is mortality. This does not validate the efficacy of a sepsis prediction system. The paper needs to be modified to reflect this differential.

Response: We completely agree with your opinion. To validate the efficacy of a sepsis definition is to determine if there are systemic responses associated with infection; however, these are very difficult to detect, and thus, many physicians simply refer to “septic” patients rather than strictly defined “sepsis.” Therefore, predicting the mortality rate, which is an important indicator in critical care patients, is way of indirectly evaluating the efficacy of a sepsis diagnostic tool. Furthermore, several studies have reported the validation of sepsis diagnostic tools using mortality as primary outcomes [1, 2]. We have clarified these points in the revised Discussion section.

3. Missing values were estimated using several different methodologies. No information is given with respect to the number of missing values or how the analysis would be different when run with only existing values. This needs to be added to the paper, as imputation of values introduces bias.

Response: The proportion of missing values is shown in S1 and S2 Figures. Two methods of imputation used: missing data at admission were replaced by normal values, while other missing data were treated using the last observation carried forward. The proportion of missing data at admission was less than 5% for all variables. All missing values at admission were imputed as normal values because the relevant tests were presumably not performed as the attending physician assumed that they would return normal results. The last observation carried forward imputation method, which is widely used for clinical trial data, was employed for missing longitudinal values. Only 4 longitudinal biomarkers (PT, lactate, TB, and bicarbonate) had more than 30% missing values. We have explained our approach for dealing with missing data in the revised Methods section.

4. The sepsis markers are measured daily throughout admission rather than at a fixed time point. How can we use such a system to predict sepsis?

Response: Most clinical data are longitudinal. Using deep learning analysis of longitudinal data is gaining increasing attention in the medical field given the potential for real-time prediction these days. In this study, this approach using longitudinal data makes it possible to perform daily predictions and allows for an early diagnosis and give the concept approach using longitudinal data. This would allow for sepsis and other outcomes to be more accurately predicted using longitudinal data collected during routine clinical care. 

Reviewer #3: Gold standard definition for sepsis noted as acute change in SOFA scores, please provide a cross tab table of patients meeting the Sepsis definition (SIRS, SEPSIS-3, etc) vs actual concurrent culture positivity. 

Response: The number of patients meeting the sepsis criteria and actual concurrent culture positivity have been described in the revised Table 1. 

That is to say I'm not surprised large burns all seem to have at least one infectious complication, but by table 2 alone and the accompanying results I'm not convinced of the temporal relation of these findings. Figure 3 makes a suggestion that there is a temporal relationship please expand and state explicitly.

Response: To address your concerns, we have provided additional data, including the timing of sepsis detection and the source of infection according to the sepsis category, as supplementary materials. Figure 3 presents the timing of the detection of each sepsis criterion from the same cohort, showing how the earliest diagnosis was made using Sepsis-3.

Additionally in a cohort of burn patients with a median TBSA of 32. I would expect multiple operative visits. Each visit can cause a systemic response similar to sepsis, how was this controlled for in the study. Is there a temporal relationship to the development of the above criteria linked to operative intervention?. Does that hold durable to predict sepsis in burn?.

Response: We completely agree that operative visits would cause a systemic response. This study compares Sepsis-1, Sepsis-3, and burn sepsis in the same cohort. Therefore, these operative visits would equally affect all sepsis criteria. In other words, all sepsis criteria were compared under the same conditions. Thus, there was no temporal relationship between the development of the sepsis criteria and the operative intervention. 

All in a very important and timely paper.

Response: Thank you for your positive comment.

 

1. Freund Y, Lemachatti N, Krastinova E, Van Laer M, Claessens YE, Avondo A, Occelli C, Feral-Pierssens AL, Truchot J, Ortega M, Carneiro B, Pernet J, Claret PG, Dami F, Bloom B, Riou B, Beaune S, (2017) Prognostic Accuracy of Sepsis-3 Criteria for In-Hospital Mortality Among Patients With Suspected Infection Presenting to the Emergency Department. JAMA 317: 301-308

2. Mak MHW, Low JK, Junnarkar SP, Huey TCW, Shelat VG, (2019) A prospective validation of Sepsis-3 guidelines in acute hepatobiliary sepsis: qSOFA lacks sensitivity and SIRS criteria lacks specificity (Cohort Study). Int J Surg 72: 71-77

---

## [Decision Letter · Decision Letter 1]

11 Oct 2022

Validation of Sepsis-3 using survival analysis and clinical evaluation of quick SOFA, SIRS, and burn-specific SIRS for sepsis in burn patients with suspected infection

PONE-D-22-15515R1

Dear Dr. Hur,

We’re pleased to inform you that your manuscript has been judged scientifically suitable for publication and will be formally accepted for publication once it meets all outstanding technical requirements.

Kind regards,

David M. Burmeister, PhD

Academic Editor

PLOS ONE

Additional Editor Comments (optional):

Reviewers' comments:

Reviewer's Responses to Questions

**Comments to the Author**

1. If the authors have adequately addressed your comments raised in a previous round of review and you feel that this manuscript is now acceptable for publication, you may indicate that here to bypass the “Comments to the Author” section, enter your conflict of interest statement in the “Confidential to Editor” section, and submit your "Accept" recommendation.

Reviewer #1: (No Response)

Reviewer #2: All comments have been addressed

Reviewer #3: All comments have been addressed

2. Is the manuscript technically sound, and do the data support the conclusions?

Reviewer #1: (No Response)

Reviewer #2: Partly

Reviewer #3: (No Response)

3. Has the statistical analysis been performed appropriately and rigorously? 

Reviewer #1: (No Response)

Reviewer #2: I Don't Know

Reviewer #3: (No Response)

4. Have the authors made all data underlying the findings in their manuscript fully available?

Reviewer #1: (No Response)

Reviewer #2: Yes

Reviewer #3: Yes

5. Is the manuscript presented in an intelligible fashion and written in standard English?

Reviewer #1: (No Response)

Reviewer #2: Yes

Reviewer #3: Yes

6. Review Comments to the Author

Reviewer #1: I would like you to please make your conclusions an recommendations more straight forward. What can you conclude based on your data and what can you recommend? how to translate this into the clinical practice?

you are missing the studies by jeschke et al working on sepsis and sepsis definitions. they looked at sepsis 3.

when did spesis occur?

Reviewer #2: The authors partially responded to my questions. The definition for sepsis diagnosis in this study does not correlate with any national or international sepsis definition.

Reviewer #3: Questions and reviews answered adequately for a retrospective review. Modifications made adress previous review

7. PLOS authors have the option to publish the peer review history of their article (what does this mean?). If published, this will include your full peer review and any attached files.

Reviewer #1: No

Reviewer #2: No

Reviewer #3: No

---

## [Editor Report · Acceptance letter]

8 Dec 2022

PONE-D-22-15515R1 

Validation of Sepsis-3 using survival analysis and clinical evaluation of quick SOFA, SIRS, and burn-specific SIRS for sepsis in burn patients with suspected infection 

Dear Dr. Hur:

I'm pleased to inform you that your manuscript has been deemed suitable for publication in PLOS ONE. Congratulations! Your manuscript is now with our production department. 

Kind regards, 

on behalf of

Dr. David M. Burmeister 

Academic Editor

PLOS ONE